# Cell-Free Screening, Production and Animal Testing of a STI-Related Chlamydial Major Outer Membrane Protein Supported in Nanolipoproteins

**DOI:** 10.3390/vaccines12111246

**Published:** 2024-11-01

**Authors:** Mariam Mohagheghi, Abisola Abisoye-Ogunniyan, Angela C. Evans, Alexander E. Peterson, Gregory A. Bude, Steven Hoang-Phou, Byron Dillon Vannest, Dominique Hall, Amy Rasley, Dina R. Weilhammer, Nicholas O. Fischer, Wei He, Beverly V. Robinson, Sukumar Pal, Anatoli Slepenkin, Luis de la Maza, Matthew A. Coleman

**Affiliations:** 1Physical and Life Sciences Directorate, Lawrence Livermore National Laboratory, Livermore, CA 94551, USAabisoyeogunn1@llnl.gov (A.A.-O.); hoangphou1@llnl.gov (S.H.-P.); vannest@uchc.edu (B.D.V.); rasley2@llnl.gov (A.R.); robinson127@llnl.gov (B.V.R.); 2Department of Pathology and Laboratory Medicine, University of California, Irvine, CA 92697, USA

**Keywords:** Chlamydia, nanodisc, cell-free expression, major outer membrane protein, membrane protein, mouse model, lipids, vaccine

## Abstract

Background: Vaccine development against Chlamydia, a prevalent sexually transmitted infection (STI), is imperative due to its global public health impact. However, significant challenges arise in the production of effective subunit vaccines based on recombinant protein antigens, particularly with membrane proteins like the Major Outer Membrane Protein (MOMP). Methods: Cell-free protein synthesis (CFPS) technology is an attractive approach to address these challenges as a method of high-throughput membrane protein and protein complex production coupled with nanolipoprotein particles (NLPs). NLPs provide a supporting scaffold while allowing easy adjuvant addition during formulation. Over the last decade, we have been working toward the production and characterization of MOMP-NLP complexes for vaccine testing. Results: The work presented here highlights the expression and biophysical analyses, including transmission electron microscopy (TEM) and dynamic light scattering (DLS), which confirm the formation and functionality of MOMP-NLP complexes for use in animal studies. Moreover, immunization studies in preclinical models compare the past and present protective efficacy of MOMP-NLP formulations, particularly when co-adjuvanted with CpG and FSL1. Conclusion: Ex vivo assessments further highlight the immunomodulatory effects of MOMP-NLP vaccinations, emphasizing their potential to elicit robust immune responses. However, further research is warranted to optimize vaccine formulations further, validate efficacy against *Chlamydia trachomatis*, and better understand the underlying mechanisms of immune response.

## 1. Introduction

*Chlamydia trachomatis* (Ct) is an increasing global public health concern that affects millions [1] as one of the most common STIs, responsible for numerous clinical manifestations [2,3]. Chlamydia infection is often asymptomatic and can result in serious lifelong health complications such as vision impairment, pelvic inflammatory disease (PID), infertility, and ectopic pregnancy [4].

There are 18 different serovars and three distinct clinical manifestations (genital, ocular, and respiratory) [5]. Serovars A-C results in serious ocular impairment by chronic conjunctivitis; Serovars D-K cause genital tract and neonatal infections [6,7]; Serovars L1-L3 manifest as an ulcerative disease of the genital area associated with genital ulcers or papules that may vary in severity [5]. Timely and effective treatment is crucial to not only alleviate symptoms but also prevent significant complications and further infection. Current treatments include antibiotics, which are generally curative but pose several limitations, including potential antibiotic resistance, re-infection, and limited cross-protection against other STIs, such as gonorrhea or syphilis [2,3]. An effective vaccine will reduce population reliance on antibiotics and slow the emergence of antibiotic resistance [2,8].

Ct belongs to the *Chlamydiaceae* family and is a gram-negative, anaerobic, obligate intracellular organism that replicates within eukaryotic cells [9]. There is a distinct biphasic lifecycle characterized by two morphological forms: metabolically inactive extracellular elementary bodies (EB) and metabolically active intracellular reticulate bodies (RB) [6,10]. Membrane proteins play a pivotal role in mediating the interactions between Chlamydia and host cells, facilitating EB entry into host cells and the subsequent differentiation into metabolically active RBs [11]. Furthermore, membrane proteins participate in host cell manipulation, nutrient acquisition, and evasion of host immune responses [9]. Understanding these dynamics provides insight into Chlamydia pathogenesis and informs the development of effective intervention strategies. *Chlamydia muridarum* (Cm), the mouse-specific analog to human Ct, is used to study immune responses due to its pathogenicity in mice and its ability to model human female genital tract infections [12]. In particular, previous studies have shown that the major outer membrane protein (MOMP) of Chlamydia has been identified as a promising vaccine candidate [13,14]. MOMP is highly conserved across Chlamydia and natively forms a trimeric structure that is membrane-bound, accounting for over 60% of the rigid, outer-membrane surface of EBs. MOMP is highly immunogenic with many B- and T-cell epitopes but is difficult to produce due to issues with structural aggregation and insolubility [15,16,17].

CFPS allows for transcription and co-translation of otherwise insoluble membrane proteins, including chlamydial membrane proteins such as MOMP (as well as polymorphic membrane, inclusion body, translocase III proteins, etc.) within nanolipoprotein particles (NLPs) [18,19]. The traditional NLP assembly method requires purified scaffold proteins, detergent-solubilized membrane proteins, and lipids for control over NLP composition, making NLPs an ideal platform for targeted and tailored vaccine design [20,21,22]. We use a faster approach that combines apolipoprotein- and membrane-protein-encoded DNA with lipids into a cell-free reaction chamber incubated at 25–30 °C for 12–18 h. This approach does not require detergents or pre-purified proteins; empty NLPs can also be made, purified, and lyophilized to support future protein production and purification reactions [23].

CFPS-produced and purified MOMP-incorporating NLPs are a promising vaccine candidate. Previous studies tested MOMP-NLP purified with a cell-free expression system [21]. MOMP-NLP vaccine formulations elicited robust humoral and cellular immune responses in preclinical mouse models, demonstrating significant induction of MOMP-specific antibodies and cytotoxic T lymphocyte (CTL) responses against Chlamydia infection [22]. Various studies have also explored recombinant MOMP vaccines with distinct delivery systems and adjuvants. For instance, a study by Sahu et al. demonstrated that PLA-PEG nanoparticles encapsulating recombinant MOMP induced robust antigen-specific Th1 and Th17 cytokine responses, alongside the proliferation of CD4+ T-cells and the development of effector T-cell phenotypes, resulting in protective immunity against genital Cm challenges in mice [24]. Additionally, a study by Tu et al. showed a MOMP-based vaccine elicited strong humoral and cellular immune responses to Ct through the generation of specific antibodies and cytotoxic T lymphocytes in a mouse model [25].

NLPs as carriers for vaccines, particularly in the context of summarizing our research utilizing past and present formulations based on MOMP-NLPs, offer insight into helping highlight formulations for the development and testing of Chlamydia vaccines. Consolidation of the current data shows the advantages related to NLP formulations include rapid production in cell-free expression, the capacity to incorporate additional components such as adjuvants, and the ability to mimic the native environment of a membrane-bound protein [23]. By incorporating adjuvants and other immunomodulatory agents, our approach with MOMP-NLPs can enhance immune efficacy against Chlamydia [21]. There is substantial potential for NLPs as a platform for membrane-bound Chlamydia antigens (including MOMP), offering a promising avenue to advance vaccine development. Here, we present a summary of new and previously published data on the use of cell-free NLP techniques for the production and testing of MOMP-NLP formulations. Importantly, we consolidate MOMP-related vaccine data from in vivo and ex vivo experiments.

## 2. Materials and Methods

### 2.1. Plasmids

Plasmids were previously described and published [21,22]. Briefly, codon-optimized sequences for the mouse ApoA1(Δ1-49), also known as Δ49ApoA1 gene, and mouse MOMP (mMOMP)gene were assembled from oligonucleotides and cloned into Nde1/BamHI-digested pVEX2.4d vector (Roche Molecular Diagnostics) via Gibson Assembly as previously described [22]. The pIVEX24.d cloned with Δ49ApoA1 gene included a His tag used for downstream nickel affinity purification. The mMOMP gene-containing plasmid does not contain a His-tag.

### 2.2. DMPC/Telodendrimer Preparation

Production and use of telodendrimer as a nanodelivery tool for membrane protein support have been described and previously published [26]. The general protocol is as follows: PEG^5000^-CA_8_ telodendrimer is stored in powder at 20 °C and is reconstituted with UltraPure DI water to a final working concentration of 2mg/mL. Preparation of small unilamellar vesicles (SUVs) from 1,2-dimyristoyl-sn-glycero-3-phosphorylcholine (DMPC) lipid were sonicated using a qSonica Q500 and one-eighth inch probe tip. The DPMC (stored in powder at −20°C) was reconstituted with UltraPure DI H2O to a concentration of 20 mg/mL and sonicated using 15 s on and 15 s off pulse intervals at 22% amplitude for 2 min with a total input energy range of 600–700 J. The sonicated lipids are centrifuged at 14,000 rcf for 1 min to remove metal contamination from the probe tip. For the DMPC/PEG^5000^-CA_8_ mixture, a 1:1 ratio of 20 mg/mL DPMC and 2 mg/mL PEG^5000^-CA_8_.

### 2.3. Cell-Free Reaction

Scalable reactions (25 µL, 50 µL, and 1 mL) were performed with the RTS 500 Proteomaster *E. coli* HY kit (Biotechrabbit GMbH, Hannover, Germany). Small-scale reactions contained the same ratio of components as large-scale reactions. Reaction components included lysate, reaction mixture, feeding mixture, amino acid mixture, and methionine, which were added sequentially per manufacturer instructions. MOMP-NLP complex is produced by the addition of 0.88 µg of delta49ApoA1 and 15 µg of mMOMP plasmid DNA to each 1 mL reaction. DMPC and telodendrimer are added in equivalent 100 µL volumes per reaction, for a total of 200 µL DMPC/telodendrimer mixture per 1 mL reaction. Reactions are incubated at 30 °C shaking at 300 rpm for 16–18 h.

### 2.4. Affinity Purification of NLP-Related Complexes

Gravity nickel affinity chromatography is used to isolate the MOMP-NLP complex from a cell-free reaction mixture. 1 mL of a slurry of cOmplete His-Tag Purification Resin (Roche Molecular Diagnostics, Pleasanton, CA, USA) was equilibrated with lysis buffer (50 mM NaH_2_PO_4_, 300 mM NaCl, pH 8.0) containing 10 mM imidazole in a 10 mL chromatography column. The 1 mL cell-free reaction was mixed with equilibrated resin and incubated at 4 °C for 1 h. Post-incubation, the sample was washed with 1 mL of 20 mM imidazole buffer six times. MOMP-NLPs were eluted in six 300 µL fractions of buffer containing 250 mM imidazole and one final elution of 300 µL in 500 mM imidazole. Elutions were analyzed downstream via SDS-PAGE to identify peak fractions, which are further pooled and dialyzed into 50 mM Tris-HCl, 300 mM NaCl, pH 7.5, and then stored at 4 °C. Protein quantifications were performed using a Qubit instrumentation according to manufacturer’s instructions (ThermoFisher Scientific, Carlsbad, CA, USA). MOMP-NLP samples for mouse studies were tested for endotoxin levels with the Endosafe-PTS (Charles River, Charleston, SC, USA) endotoxin testing system based on Limulus amebocyte lysate. MOMP-NLP preparations had an average endotoxin between 1000 and 1500 endotoxin units per milliliter.

### 2.5. Size Exclusion Chromatography (SEC)

NLPs are purified by SEC (Superdex 200, 3.2/300 GL column, GE Healthcare, Darmstadt, Germany). SEC was run at a flow rate of 0.2 mL/min in PBS buffer with 0.25% PEG 2000.

### 2.6. SDS-PAGE

1–5 µL aliquots of eluted MOMP-NLPs were mixed with 4× NuPAGE lithium dodecyl sulfate sample buffer and 10× NuPAGE sample reducing agent (Life Technologies, Carlsbad, CA, USA), heat-denatured at 95 °C for 5 min and loaded onto a 4–12% gradient premade 1.0 mM Bis-Tris gel (Life Technologies) with molecular weight standard SeeBlue Plus2 (Life Technologies). The running buffer was 1× MES-SDS (Life Technologies). Samples were run at 200 V for 35 min. Gels were stained with SYPRO Ruby protein gel stain (Life Technologies) per the manufacturer’s instructions and imaged using LI-COR Odyssey imager. Protein bands were quantified using Image Studio V2.0 software via densitometry.

### 2.7. Western Blot Analyses

SDS-Page gel was run as previously described [22]. Briefly, the gel was transferred to the PVDF membrane after running using the iBlot 2 Dry Blotting System (Invitrogen, Waltham, MA, USA) for 10 min per manufacturer instructions. The membrane was then incubated at 4 °C for 1 h in LI-COR blocking buffer, followed by incubation in primary antibody (1:100 for mAb40) with LI-COR blocking buffer and 0.2% Tween 20 overnight at 4 °C. The following day, the membrane was then washed for 5 min in 1× PBS-T (0.2% Tween 20, 1× PBS at pH 7.4) four times. The membrane was incubated and protected from light for 1 h in a secondary IR800 antibody (1:10,000 dilution) with blocking buffer and 0.2% Tween 20 at room temperature. Subsequently, 5-min wash steps with 1× PBS-T were repeated four times. Afterwards, the membrane was washed with 1× PBS at room temperature to remove any residual detergent. Membranes were then imaged using LI-COR Odyssey imager.

### 2.8. Dynamic Light Scattering (DLS)

Dynamic light scattering measurements of the NLP size were performed on a Zetasizer Nano ZS90 (Malvern Instruments, Malvern, UK) following the manufacturer’s protocols. Each data point represents an average of at least 10 individual runs.

### 2.9. Animal Vaccinations for Immunological Studies

Four to five-week-old female BALB/c (H-2d) mice (Charles River Laboratories; Wilmington, MA, USA) were housed at the Lawrence Livermore National Laboratory (LLNL) Vivarium. Animal protocols used were approved by the LLNL IACUC. The adjuvants CpG-1826 (TriLink, San Diego, CA, USA; 10 g/mouse/immunization) and FSL1 were directly mixed with single antigens (MOMP-NLP or MOMP at 10 μg of each antigen/mouse/immunization) and different adjuvant combinations. Groups of 5 mice were immunized in a prime-boost-boost regimen 3 weeks apart via the intramuscular (i.m.) route in the quadriceps muscle. To determine the cell-mediated immune responses, all mice are euthanized 7–10 days post-last boost, and tissue of interest is harvested. All animal experiments were replicated at least once.

### 2.10. RNA Extraction and Gene Expression Analysis

A fraction of the mouse spleen is collected at 7–10 days post-last boost and stored in RNAlater (Thermo Fisher, Waltham, MA, USA). Total RNA is isolated using the column-based nucleic acid purification kit (RNeasy, Qiagen, Hilden, Germany), strictly according to the manufacturer’s protocol. Spleen samples were placed in reinforced polypropylene tubes containing ceramic beads (Omni Bead Ruptor tubes) and the appropriate lysis buffer. Samples were homogenized and subjected to a multi-step washing and centrifugation protocol. RNA purity and final concentration are determined, followed by cDNA synthesis by reverse transcription (RT). PCR analysis was used to screen for changes in gene expression related to Th1 and Th2 responses. For this, the cDNA of 5 animals in each experimental group were pooled and mixed with the PCR master mix buffer (Rt^2^ Syber Green ROX qPCR Primer Assay, Qiagen) and analyzed on specific array plates (RT^2^ Profiler™ PCR Array Mouse Th1 and Th2 Responses- GeneGlobe ID—PAMM-034Z). PCR-array analysis was performed using a ViiA™7 Real-Time PCR System (Life Technologies), and the results were assessed with the 12 K Flex QuantStudioTM (https://www.thermofisher.com/us/en/home/technical-resources/software-downloads/quantstudio-12k-flex-real-time-pcr-system.html, accessed on 21 September 2024) software (Applied Biosystems, Waltham, MA, USA). The expression of all genes of interest is presented as fold change relative to sham. Further pathway-focused gene analyses were performed based on the “Enrichr” gene enrichment analysis online tool [27,28,29].

### 2.11. Ex Vivo Splenocyte Restimulation and Multiplex Cytokine/Chemokine Array

Spleens from vaccinated mice for immunological studies are collected at 7–10 days post-last boost and processed into single cells. Splenocytes are seeded in 24 well plates at 2 × 10^6^ cells per well in 0.5 mL of RPMI media with glutamine, supplemented with 10% fetal bovine serum (FBS) and 100 U/mL penicillin-streptomycin (Invitrogen Life Technologies). Splenocytes from all vaccinated groups including the PBS vaccinated group are restimulated with MOMP protein at 10 μg/mL while only splenocytes from the PBS group are stimulated with PMA as a control. This is followed by a 72-h incubation, after which the conditioned media is collected, and the same volume of sterile PBS is added for a two-fold dilution. The conditioned media were profiled using the 32-plex discovery assay (mouse cytokine 32-plex discovery assay, Cat. No: MD32, Eve Technologies Corp., Alberta, Canada). The 32-plex discovery assay consisted of Eotaxin, G-CSF, GM-CSF, IFNγ, IL-1α, IL-1β, IL-2, IL-3, IL-4, IL-5, IL-6, IL-7, IL-9, IL-10, IL-12 (p40), IL-12 (p70), IL-13, IL-15, IL-17, IP-10, KC, LIF, LIX, MCP-1, M-CSF, MIG, MIP-1α, MIP-1β, MIP-2, RANTES, TNFα, and VEGF.

### 2.12. Challenge Studies

Four to five-week-old female BALB/c (H-2d) mice (Charles River Laboratories; Wilmington, MA, USA) were housed at the University of California, Irvine, Vivarium. The University of California, Irvine IACUC approved all animal protocols. The adjuvants CpG-1826 (TriLink, San Diego, CA, USA; 10 μg/mouse/immunization) and Montanide ISA 720 VG (SEPPIC Inc., Fairfield, NJ, USA; 70% of total vaccine volume), FSL1 were directly mixed with single antigens (MOMP-NLP or MOMP: 10–20 μg of each antigen/mouse/immunization) and antigens combinations (10 μg of each antigen/mouse/immunization). Groups of 5 to 9 mice were immunized twice by the intramuscular (i.m.) route in the quadriceps muscle at a 4-week interval. Immunization controls included EB vaccinations, adjuvant control groups immunized with CpG-1826, and Montanide ISA 720 VG in phosphate-buffered saline (PBS). Four weeks after the last immunization, mice were challenged intranasally (i.n.) with 10^4^ IFU of Cm. All animal experiments were replicated once.

### 2.13. Statistical Analyses

Statistical analyses of secreted cytokines were performed using GraphPad Prism (version 10.1.1) software, and comparison between groups was performed using a two-tailed nonparametric Mann–Whitney *t*-test. Parametric and non-parametric statistical tests were used as follows for the animal challenge study. The Student’s *t*-test was employed to evaluate changes in body weight on day 10 p.c., lungs’ weights, and amounts of IFN in lungs supernatants. Repeated measures ANOVA was used to compare changes in mean body weight over the 10 days of observation following the Cm i.n. challenge. The Mann–Whitney U-Test was used to compare antibody titers, levels of IFN- and IL-4 in T-cell supernatants, and the number of Cm IFU in the lungs. Values below the limit of detection (BLD) were assigned the value of the BLD, as described by Beal. A *p*-value of <0.05 was considered significant while *p*-values of <0.1 were considered approaching significance.

## 3. Results

### 3.1. Nanolipoproteins (NLPs) Are a Tool for Studying Membrane Bound Proteins

NLP assemblies can be assembled using cell-free methods with or without the membrane proteins to support both in vitro and in vivo studies (Figure 1). An NLP is an 8–25 nm membrane-mimetic disc composed of phospholipids and a stabilizing scaffold protein, such as apolipoproteins (Figure 1A). Combining the apolipoprotein-encoded plasmid and lipid into a cell-free reaction yields “empty” NLPs without an associated membrane protein (Figure 1B). Cargo, such as a desired membrane protein, can be efficiently loaded by lipids and plasmids encoding the scaffold and membrane proteins in the same cell-free reaction. This reaction results in the incorporation of the full-length membrane protein into the assembled NLP (Figure 1C). Yields range from μg’s to mg’s of membrane protein containing NLPs, depending upon factors such as scale of reaction, plasmid backbones for protein components, and the associated membrane proteins.

### 3.2. Cell-Free Screening Enables Rapid Lipid Testing and Optimization

To optimize the solubility of MOMP in NLPs, we screened 12 different lipid/polymer NLP formulations. Each condition was run as a 25 µL *E. coli*-based cell-free reaction with the scaffold protein ApoA1. Specifically, we screened EggPC, DOPE, DOPC, and DMPC lipids for soluble MOMP after co-translation with ApoA1 and incorporation of fluorescently labeled amino acids (Figure 2A). We also combined these lipids with telodendrimer PEG^5000^-CA_8_ (Telo) in our screen, which has previously been shown to increase complex solubility [22]. Out of the four individually tested lipids, DMPC performed the best, indicated by denser bands at 40kDa representing MOMP-encapsulated NLP compared to EggPC, DOPE, and DOPC. Notably, the addition of telodendrimer PEG^5000^-CA_8_ (Telo) to DMPC reactions improved the total MOMP-NLP complex and improved solubility. Furthermore, other conditions of the cell-free reaction, such as reaction volume, membrane/scaffold protein plasmids, reaction time, and temperature, can be systematically investigated to optimize the production of soluble membrane-protein-containing NLP. For example, in a 100 µL *E. coli*-based cell-free reaction, we demonstrated optimization of the NLP scaffold protein (Figure 2B). The addition of the ApoA1 scaffold protein plasmid results in stronger MOMP-NLP bands in comparison to the ApoE4 scaffold protein plasmid. However, omitting a scaffold protein entirely dramatically reduces MOMP-NLP solubility to nearly undetectable levels compared to ApoA1 and ApoE4 scaffold proteins. Taken together with the results from the lipid screen performed previously, we identified the most effective formulation of soluble MOMP-NLP complex is ApoA1 scaffold protein with DMPC and Telo combinations for supporting formulation scale-up, purification and characterization to support in vivo vaccine-related studies.

### 3.3. Cell-Free Production and Purification of MOMP-NLP Protein Complex

We first reported MOMP solubilization and characterization in NLPS using cell-free techniques in the Wei et al. publication in 2016. In addition to serving as a viable method for rapid formulation screening, cell-free reactions are scalable to produce sufficient yields of MOMP-NLP for in vivo immune studies [30]. For cell-free production of MOMP-NLP, we increased the *E. coli*-based reaction volume scale to a 1–2 mL reaction scale. A detailed protocol for cell-free MOMP-NLP production can be found in the materials and methods section. Briefly, self-assembly of MOMP-NLP complexes is carried out in a cell-free reaction by combining MOMP and ApoA1 protein-encoded plasmids with lipids and telodendrimers in an *E. coli*-based cell lysate reaction mix (Figure 3A). Affinity purification with Ni^2+^-NTA-resin results in two distinct bands on the SDS-PAGE gel, which confirms two proteins in the complex, MOMP and ApoA1. Unlike the ApoA1 protein, the translated MOMP protein does not contain a His-tag, indicating MOMP is incorporated into the NLP and subsequently co-purified as part of the membrane-protein NLP complex. Quantification and densitometry of affinity purified and dialyzed MOMP-NLPs were performed and indicate MOMP protein to ApoA1 scaffold protein ratios range from 1:1 to 3:1. A typical 1 mL reaction yields 1.5–2 mgs of total protein per mL when including both MOMP and ApoA1 proteins (Figure 3B). Following dialysis in Tris-based buffers, yields range from 300 to 500 µg of pure MOMP membrane protein in the MOMP-NLP complex for downstream use, thus representing 30–50 potential doses of vaccine per mL of reaction volume. Characterization of purified and dialyzed MOMP-NLP complexes by Western blot with the MOMP-specific mAb40 antibody (Figure 3C), which targets a linear epitope located on the extracellular domain of the MOMP protein [21] confirms the protein co-purified with NLPs is MOMP. Taken together, these results indicate that cell-free production of MOMP-NLPs is readily purifiable via affinity chromatography and may be subsequently used in downstream biophysical characterization, in vitro, and in vivo assays.

### 3.4. Characterization of the MOMP-NLP Complex

Biophysical methods are used to ensure the formation of functional MOMP-NLP oligomeric complexes. Negative stain cryo-TEM image analysis of the peak MOMP-NLP elution fractions (Figure 4A) confirms the association and insertion of MOMP protein into the NLP. Multiple MOMP proteins can be incorporated in one nanodisc, representing the native trimer formation of MOMP. Additionally, dynamic light scattering (DLS) also supports the insertion of MOMP protein into NLPs. Empty NLPs (not shown) measure at approximately 10 nanometers, whereas MOMP-NLPs measure closer to 45 + standard deviation nm (Figure 4B) [22]. Electrophysiology assays were used to confirm that the MOMP-NLP protein forms a functionally active porin within lipid bilayers (Figure 4C). The electrophysiology current trace indicates an open pore (top), as shown by a 10 pA jump. A flat trace (bottom) indicates a lack of pore function either due to a lack of the inserted porin or from a closed pore (Figure 4D). Additional characterization by size-exclusion chromatography (SEC) analysis of peak elution fractions confirmed the homogeneity of MOMP-NLP complexes [22]. In previous experiments, the MOMP-NLP complex elutes at approximately 7 min and indicates the complex can be separated from free proteins or lipid aggregates [22].

### 3.5. Effectiveness of NLP Encapsulated MOMP-Based Vaccine Formulations Administered Through Various Routes and Immunization Schedules

To test the effectiveness of our NLP-encapsulated MOMP-based vaccine formulations, we explored several routes of vaccine administration (intranasal- IN and intramuscular- IM vaccinations) with both prime and prime-boost vaccination regimens in BALB/c female mouse strains (Figure 5 and Figure 6). To complement these studies, we also explored a prime-boost-boost vaccination schedule and used ex vivo splenocyte cultures to assess the immune-relevant pathways (Figure 7). The table (Figure 8) summarizes how rMOMP and MOMP-NLP vaccine formulations, tested with different adjuvant combinations and vaccination routes in a Cm challenge study, effectively enhance immune responses against Chlamydia infections.

In vivo studies relied on intranasal challenge studies with Cm four weeks after the last vaccination, followed by tissue harvest 10-days post-challenge (Figure 5A and Figure 6A). Systemic and local disease burden following the intranasal Cm challenge of all vaccinated mice were evaluated by measuring body weight, lung weight, and the log IFU of Cm recovered from the lungs for all the vaccination schedules. Mice vaccinated using the IN/IM prime-boost regimen with recombinant MOMP-NLP adjuvanted with 5 µg CpG and 1 µg FSL1 showed over 1 log fold reduction of Cm IFUs recovered from the lungs (at a mean of 5.8 log IFU, mean) when compared to mice vaccinated with the same schedule and formulation but with a lower dose of CpG, 1 µg (7.1 log IFU, mean), and almost a 3 log fold reduction of Cm IFUs when compared to the IFUs recovered from the lungs of mice vaccinated with a sham control, PBS (8.6 log IFU, mean) (Figure 5B). Interestingly, our single adjuvanted recombinant MOMP-NLP formulation with only 5 µg of CpG showed the same level of protection (7.1 log IFU, mean) as the formulation with the lower dose of CpG (1 µg) and FSL1 at 1 µg (7.1 log IFU, mean) (Figure 5B). Additionally, mice vaccinated with MOMP-NLP adjuvanted with 5 µg CpG and 1 µg FSL1 showed a significantly lower change in body weight and reduced lung weight when compared to the PBS-vaccinated mice (Figure 5C). All mice vaccinated with either a MOMP or MOMP-NLP-based formulation showed a significant reduction of Cm IFUs recovered from their lungs when compared to mice vaccinated with PBS (Figure 5D).

Although a combined IN/IM vaccination scheme shows promising results for a protective vaccine, we also wanted to understand the effectiveness of our MOMP-NLP antigens administered solely through the IM vaccination route. For this vaccination scheme, we tested our MOMP-NLPs along with varied concentrations of CpG and FSL1 adjuvants in BALB/c female mice with an intramuscular (IM) prime vaccination followed by an intranasal challenge with Cm (Figure 6A). The MOMP-NLP antigen adjuvanted with 5 µg CpG and 10 µg FSL1 showed the highest level of protection with a mean of 7.5 log IFU of Cm recovered from the mice lungs compared to a mean of 9.06 log IFU for the formulation with lower doses of the adjuvants (1 µg CpG and 1 µg FSL1) (Figure 6B). Although not significant, the change in mice’s body weight and lung weight were lower for the formulation with 5 µg CpG and 10 µg FSL1 when compared to formulations with other concentrations of CpG and FSL1 tested (Figure 6C,D).

Lastly, we aimed to characterize how MOMP-NLP antigens and various adjuvant combinations interact with and induce the immune system to potentially elucidate the currently unknown correlates of protection against Chlamydia infections. To accomplish this, we injected MOMP-NLPs adjuvanted with 3 µg CpG and 2 µg FSL1 into Balb/c female mice following a prime-boost-boost vaccination regimen without intranasal challenges (Figure 7A). Gene expression of pooled spleen samples of MOMP-NLP + CpG + FSL1 vaccinated mice relative to pooled spleens of PBS vaccinated mice showed overexpression of Th1 genes- IFNγ, IL2, IL2RA, IL4RA, and Th2 genes- IL5, CCL7, CSF2, GATA3, IL7R, and TNFRSF8. These genes are involved in the activation of several pathways, including the regulation of the transcription factor- nuclear factor of activated T cells (NFAT), cytokine-cytokine receptor interaction, and Th1/Th2 differentiation genes (Figure 7B). Concurrently, IL1R1 and IL25, genes involved in the mediation of cytokine-induced immune and inflammatory responses and the induction of NF-kappaB activation, respectively, were downregulated (Figure 7C). Th1 and Th17 cytokines, IFNγ, CCL2, and IL17, were significantly secreted in the conditioned media of splenocytes from MOMP-NLP + CpG + FSL1 vaccinated mice cultured ex vivo and restimulated with recombinant MOMP compared with PBS vaccinated mice splenocytes stimulated with MOMP and Phorbol myristate acetate (PMA) as controls (Figure 7D). The table (Figure 8) summarizes the rMOMP and MOMP-NLP formulations tested with various adjuvant combinations and several routes of vaccination, all evaluated in a challenge study with Cm. Protection is determined by the 2-fold IFU log mean difference upon comparison to EB values, with the IFU log mean between 1 and 2 log difference classified as “semi-protective”. These findings provide a comprehensive overview of the protective efficacy of our vaccine formulations, demonstrating their effectiveness in a challenging setting, and underscore the potential of optimized adjuvant combinations in enhancing immune responses against Chlamydia infections.

## 4. Discussion

Combining the NLP technology with an *E. coli* cell-free expression system aids rapid membrane protein synthesis that maintains the native oligomeric formation necessary for effective immunogenicity [31] and is a powerful tool for the development of vaccines based on membrane-bound proteins that represent antigens derived from infectious agents. NLPs combined with cell-free expression systems [19,32,33] provide a rapid and flexible approach to producing a wide range of difficult-to-produce proteins in a format that allows for quick screening of multiple conditions to optimize antigen solubility [34]. The ability to couple the process to vaccine studies further highlights the capability of the NLP to act as a delivery vehicle, another attractive feature of this technology.

Within this paper, we present a more complete view of the production of the MOMP-NLP complex [22] using various adjuvant formulations for in vivo and ex vivo studies. Previous in vivo studies have demonstrated that the native MOMP structure is crucial to elicit a systemic immune response [21,22]. The combination of MOMP and apolipoprotein plasmid-DNA with lipid and Telo rapidly produces the MOMP-NLP complex. Previous TEM imaging studies have found that the addition of the Telo plus lipids can also serve as membrane protein support, which forms a unique type of nanodisc [22] and may aid in solubilizing lipids in the formation of MOMP-NLP [22]. It is also hypothesized that the PEGylated tail minimizes MOMP and MOMP-NLP interactions, thus reducing protein aggregation and retaining water solubility. The MOMP-NLP complex and its oligomeric formation are confirmed through several quality control checks and structural analyses, including size exclusion chromatography (SEC), TEM, SDS-PAGE, and dynamic light scattering [21,22,30]. Previous reports indicate that native trimeric MOMP is the optimal antigen to induce a protective immune response [35,36]. Electrophysiology results show that the expression of MOMP in NLPs forms a complex with MOMP, potentially with a percentage of the protein presenting as a functional trimeric channel, allowing it to oligomerize in a fashion similar to native MOMP [37]. However, these previous studies did not use native MOMP as a control in electrophysiology experiments.

Protective immunity involving both cellular and humoral immune responses following vaccination is the goal of vaccine development for infectious diseases [38]. Animal studies have demonstrated that recombinant MOMP can elicit long-lasting and protective antibody responses specific to Cm [39]. Hence, the expression of MOMP in NLPs presents a promising vaccine delivery platform that has the potential to enhance targeted antigen presentation and the colocalization of different adjuvants with MOMP for robust immunogenicity [40]. We previously demonstrated that mice vaccinated via the IN/IM prime-boost regimen with MOMP-NLP and CpG and FSL1 as adjuvants elicited partial protection after the intranasal challenge with Cm [21]. In this study, we tested different concentrations of CpG in formulation with recombinant MOMP-NLP and FSL1 and explored different routes of vaccinations. Our formulation with the higher dose of CpG administered using the IN/IM prime-boost regimen showed significantly less Cm IFUs in the mice lungs and minimal change in body weight, indicating a potent formulation resulting in the improvement of the quality and quantity of immune responses [41,42] to our recombinant MOMP-NLP formulation, generated by co-adjuvanting with CpG and FSL.

While the MOMP-NLP formulation shows promise in generating strong immune responses, it is important to contextualize its potential within the broader landscapes of our Chlamydia vaccine candidates. Live attenuated vaccines have been explored due to their ability to mimic natural infection and stimulate a broad and robust immune response, including both humoral and cellular immunity [43]. However, concerns about safety, especially the potential for reversion to a virulent form, have limited their advancement in clinical trials. Therefore, epitope-based subunit vaccines have garnered attention and subsequently demonstrated their ability to elicit targeted immune responses by focusing on conserved regions of MOMP, providing cross-serovar protection [44]. However, such formulations may lack the structural complexity needed for broad and long-lasting immunity. Full-length MOMP vaccines offer an alternative, complete antigenic profile, enhancing immune responses, but face significant challenges in solubilization and stability [45]. Our MOMP-NLP formulation attempts to overcome these difficulties by maintaining the structural integrity of full-length MOMP and improving solubility via the nanolipoprotein platform. This approach could offer improved immune responses without the safety risks of live attenuated vaccines while potentially providing better protective efficacy [30].

To understand the immunogenicity of our *E. coli* cell-free expressed MOMP-NLP complex in combination with CpG and FSL1, we utilized unchallenged, IM prime-boost-boost vaccinated mice and measured the gene expression profiles and ex vivo cytokine and chemokine secretion from spleen RNA and splenocytes, respectively. These IM vaccinations were three weeks apart, and tissue harvest was carried out 7–10 days after the last boost. The overexpressed genes induced by MOMP-NLP adjuvanted with CpG and FSL1 are shown to elicit, with high significance, the regulation of NFAT (Nuclear Factor of Activated T-cells) transcription factors. NFAT transcription factors are known for playing a crucial role in the regulation of genes involved in T-cell activation, proliferation, and differentiation and are activated upon T-cell receptor stimulation, ultimately influencing immune responses, including inflammation, immunity, and tolerance [43,44]. Cytokine-cytokine receptor interactions were also activated and are known for their crucial role in the induction of immune responses by mediating communication between cells of the immune system and other tissues [46,47]. Furthermore, we see an activation of the Th1/Th2 differentiation pathway, where naïve CD4+ T helper cells are polarized to either Th1 or Th2 cells for the orchestration of appropriate immune responses against pathogens and antigens [48,49]. The two genes downregulated with MOMP-NLP adjuvanted with CpG and FSL1 vaccination, IL1R1 and IL25, are involved in cytokine-cytokine receptor interactions as well. ILIR1 is a mediator of many cytokine-induced immune and inflammatory responses, while IL25 has been shown to be associated with type 2 immune responses responsible for the development of allergic diseases [50,51,52,53].

*E. coli* cell-free expression is a powerful technique that allows for rapid screening and optimization of Chlamydial membrane proteins that are difficult to produce. The technology can be coupled with the NLP platform to increase membrane protein solubility and yield. This allows for the formation of the functional, multimeric MOMP-NLP complex whose structure and formulation characteristics can be analyzed using biochemical techniques. Our approach focuses on techniques such SDS-PAGE, western blots, TEM, DLS, and electrophysiology. The NLP-based approach provides highly purified and behaved material for characterization and utility in animal studies. Overall, we have been able to demonstrate that Cm MOMP formulated in a bio-mimetic nanoparticle is malleable for the colocalization of different adjuvants that can enhance the robust, protective, and prolonged immune responses against Chlamydia infections.

While our past and present studies have demonstrated the efficacy and immunogenicity of our MOMP-NLP-based formulations in preclinical models of Chlamydia infection, there are several aspects that remain unexplored. Firstly, there is a need to directly compare the folding and functional properties of our MOMP-NLP complex to isolated native MOMP protein. Although our characterization assays (SDS-PAGE, western blotting, TEM imaging, and electrophysiology) suggest the formation of a functional oligomeric complex, further structure analyses, such as X-ray crystallography or cryo-electron microscopy, are necessary to confirm the folded state of MOMP within the NLP. Additional investigation into the dose-response relationship of both MOMP and different adjuvants tested is also crucial. Identifying the optimal dose of MOMP and adjuvants is key for maximizing vaccine efficacy while minimizing potential adverse effects. Previous studies using recombinant MOMP formulated with montanide have shown protective immune responses in the genital tract, reducing inflammation and the number of Chlamydia-infected fetuses [54]. This highlights the potential of full-length MOMP in eliciting protection in clinically relevant models. However, our MOMP-NLP formulation, which utilizes recombinant MOMP, has identified optimized antigen conditions to include antigen concentration and adjuvant amounts. However, new formulations still need to be tested in genital tract challenge studies. Given the importance of this anatomical site in Chlamydia infection, future studies should focus on evaluating the efficacy of MOMP-NLP in such models to more thoroughly assess its protective capabilities. Furthermore, while our experiments have Cm MOMP as the antigen, translating these findings to Ct is essential. Despite conserved regions, further studies are required to validate the immunogenicity and efficacy of our vaccine formulations against Ct infection. Overall, addressing these aspects will provide valuable insights into the development and optimization of MOMP-NLP-based vaccines for Chlamydia infection.

## 5. Conclusions

Presenting comprehensive and complete data on the generation of NLP protein complexes is crucial for evaluating the potential and necessity of further exploration into additional chlamydial antigens and adjuvants that could enhance MOMP’s efficacy as a subunit vaccine. The consolidated findings provide a unique dataset on the screening and production pipeline for MOMP-NLP vaccine development, potentially paving the way for other STI-related vaccines that target membrane-bound antigens. The NLP platform serves as an optimal tool for the co-delivery of various lipidic adjuvant combinations. Notably, MOMP-NLP formulations with CpG and FSL1 demonstrated protective effects and identified key immunological correlates essential for understanding protection in mouse models. Continued research is imperative to optimize adjuvant dosages and combinations, aiming to improve protection and reduce disease burden. A deeper understanding and strategic use of the mechanisms of action of these adjuvants, whether used individually or in combination, will be crucial in achieving the desired cellular and humoral immune responses necessary for the development of a human chlamydial vaccine.

## Figures and Tables

**Figure 1 vaccines-12-01246-f001:**
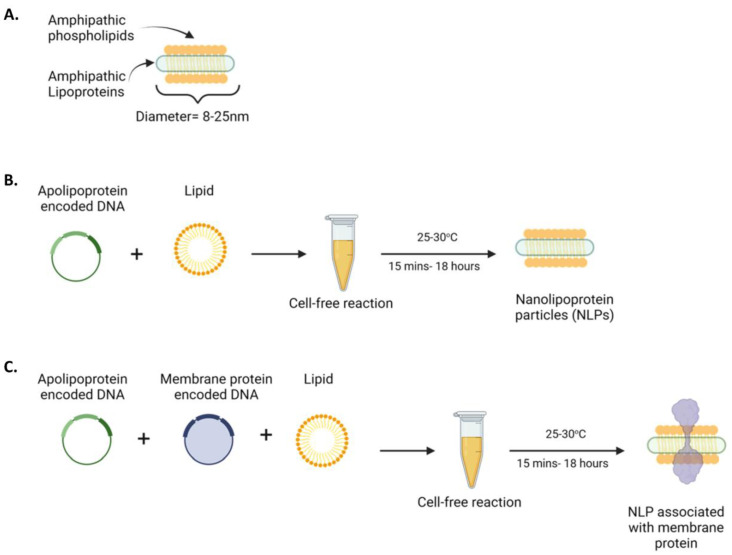
Nanolipoprotein particles are a tool for studying membrane-bound proteins. (**A**) Nanolipoprotein particles, NLPs (8–25 nm disc-shaped particles), are formed by the spontaneous assembly of phospholipids into a bilayer stabilized by an apolipoprotein scaffold protein. (**B**) Cell-free approach for NLP production without detergents or pre-purified proteins. (**C**) Cell-free approach for full-length membrane protein expression encapsulated in NLP.

**Figure 2 vaccines-12-01246-f002:**
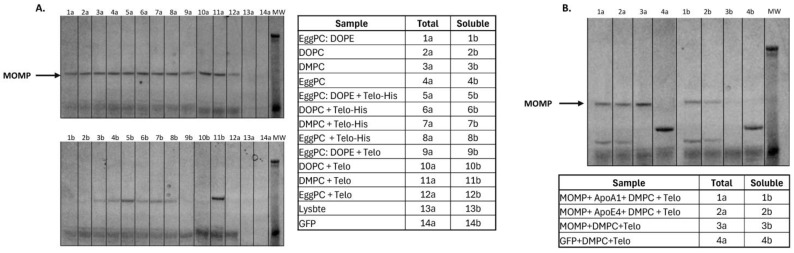
Cell-free screening allows for rapid, large-scale condition testing and optimization. (**A**) SDS—PAGE image of 25 µL MOMP protein reactions with various lipids and telodendrimer to identify a condition that optimizes solubility. (**B**) SDS page of 100 µL MOMP protein reactions with apolipoprotein scaffold and various lipid/telodendrimer combinations demonstrates the scalability of the cell-free techniques. M = Molecular weight marker.

**Figure 3 vaccines-12-01246-f003:**
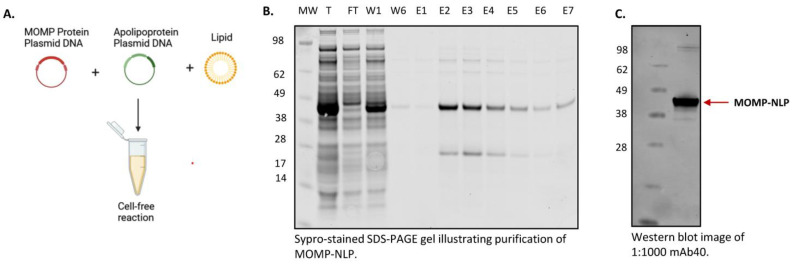
Cell-free production of MOMP-NLP protein complex for mouse studies. (**A**) Cell-free expression of MOMP-NLP showing components including MOMP DNA, Δ49ApoA1 DNA, and pre-prepared DMPC/telodendrimer lipid in a cell-free reaction chamber. (**B**) SDS-PAGE Sypro-stained image showing cell-free produced MOMP-NLP purified using Nickel bead gravity column (molecular weight (MW) marker—SeeBlue Plus2, Total protein (T), flow-through (FT)—MOMP protein not associated with an NLP, two of six washes (W)—to purify protein of interest, and seven elutions (E1–E7)—to recover MOMP-NLP. (**C**) Western blot micrograph showing cell-free produced MOMP using mAb40, a primary antibody against MOMP.

**Figure 4 vaccines-12-01246-f004:**
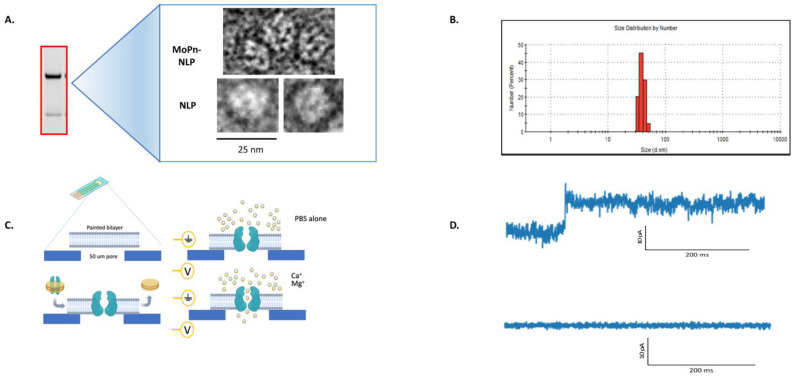
Imaging and electrophysiology techniques confirm the formation of the MOMP-NLP complex. (**A**) Negative stain cryoEM of peak fraction of eluted, soluble MOMP-NLP complex (from Figure 3B) illustrates the circular disk-like shape of Cm MOMP-NLP and empty NLP. (**B**) Dynamic Light Scattering (DLS) measurement further confirms the formation of the MOMP-NLP complex. (**C**) The MOMP protein is active as a porin, as shown by electrophysiology. Single channel conductance assay of MOMP-NLPs in fixed bilayers at a fixed voltage, standard 200 mV. (**D**) Electrophysiology additionally confirms MOMP-porin is active; the top trace shows 10 picoamp jumps to open confirmation of porin. A flat line indicates no porin activity and baseline membrane bilayer reading value.

**Figure 5 vaccines-12-01246-f005:**
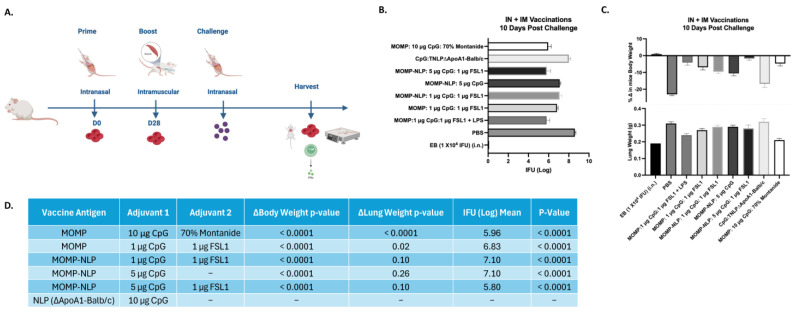
Systemic and local disease burden following intranasal Cm challenge of mice vaccinated via IN/IM prime-boost regimen with recombinant MOMP-NLP and adjuvanted with CpG and FSL1. (**A**) Experimental schematic showing intranasal prime and intramuscular boost vaccinations, i.n. challenge and tissue harvest. (**B**) Log IFU of Cm recovered from mice lungs 10 days post-challenge. (**C**) Change in mice body weight and lung weight (g) at 10 days post i.n. challenge with Cm. (**D**) A summary chart shows vaccine antigens, adjuvants, and *p*-values.

**Figure 6 vaccines-12-01246-f006:**
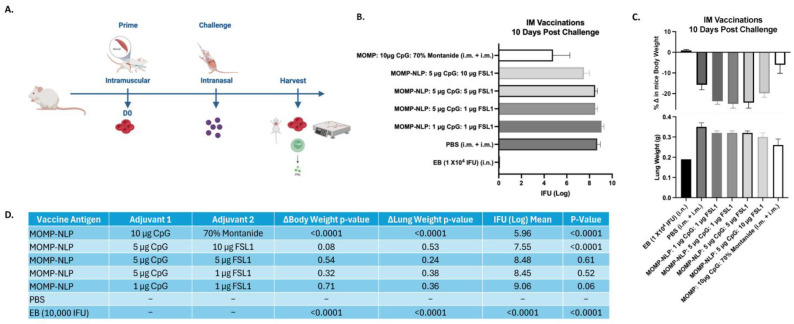
Systemic and local disease burden following intranasal Cm challenge of mice vaccinated via IM prime regimen with recombinant MOMP-NLP and adjuvanted with CpG and FSL1. (**A**) Experimental schematic showing intramuscular prime vaccinations, i.n. challenge and tissue harvest. (**B**) Log IFU of Cm recovered from mice lungs 10 days post-challenge. (**C**) Change in mice body weight and lung weight (g) at 10 days post i.n. challenge with Cm. (**D**) A summary chart shows vaccine antigens, adjuvants, and *p*-values.

**Figure 7 vaccines-12-01246-f007:**
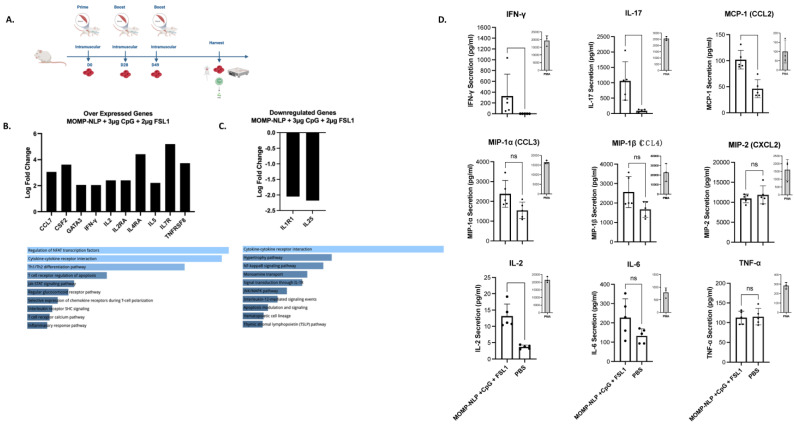
Gene and cytokine expression following prime-boost-boost mice vaccinations with recombinant MOMP-NLP adjuvanted with CpG and FSL1. (**A**) Experimental schematic showing an intramuscular prime and two intramuscular boost vaccinations followed by tissue harvest. (**B**) Overexpressed genes with greater than or equal to 2-fold change value of vaccinated mice relative to sham (PBS) vaccinated mice and respective activated pathways, ranked by *p*-value. (**C**) Illustrates the downregulated genes. For these experiments, cDNA samples were obtained from pooled spleen RNA samples of 5 different animals per group and *p*-value ranked pathways/processes generated by NCATS BioPlanet, Enrichr. (**D**) Secreted cytokines following stimulation of splenocytes 7–10 days after boost with MOMP.

**Figure 8 vaccines-12-01246-f008:**
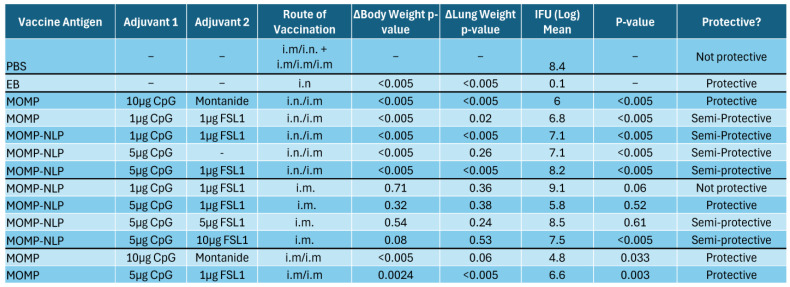
Several Cm rMOMP and MOMP-NLP vaccine formulations offer protection against Chlamydia infections. A chart summarizing rMOMP and MOMP-NLP formulations tested with various adjuvant combinations and several routes of vaccinations. Protection was determined by a 2-fold IFU log mean difference upon comparison to EB values. IFU log means between 1 and 2 log differences classified as “semi-protective”.

## Data Availability

Data is contained within the article.

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
