# Peer review of "Cell-Free Screening, Production and Animal Testing of a STI-Related Chlamydial Major Outer Membrane Protein Supported in Nanolipoproteins"

_vaccines, 2024, doi:10.3390/vaccines12111246_

Round 1
Reviewer 1 Report
Comments and Suggestions for Authors
The article "Cell-free screening, production and animal testing of the major outer membrane protein of chlamydia associated with STIs supported by nanolipoproteins" presents a modern approach to obtaining vaccines containing membrane-bound proteins of the pathogen. The authors have developed a vaccine against the membrane protein of a model object - Chlamydia muridarum, which causes disease in mice. Chlamydia trachomatis, which affects humans, is the causative agent of a sexually transmitted disease and a concern for the health care system.
In the Introduction, the authors substantiate the need to create a vaccine against Chlamydia trachomatis and the adequacy of using Chlamydia muridarium as an experimental model. They provided a full complex of experimental data in the manuscript, including previously obtained ones, demonstrating the entire cycle of vaccine production, including testing its effectiveness in vivo. Such a presentation of the material is justified, since we are talking about a SYSTEM of an effective approach to developing vaccines using membrane proteins of the pathogen.
The Materials and Methods section provides a full description of the methods and includes necessary references to publications. The information in this section is sufficient to reproduce the study.
The Results section clearly describes all steps of the work in sufficient details. Diagrams depicting the sequential steps of the MOMP-NLP complex assembly visualize this important part of the work, and data of lipid screening explain the choice of the most effective formulation of the soluble MOMP-NLP complex: ApoA1 scaffold protein with combinations of DMPC and Telo to support scaling, purification and characterization for in vivo vaccine studies. To understand the work, it was important to explain the preparation of the MOMP-NLP protein complex in a cell-free system, and the authors have successfully done this. All research results are presented in detail and well illustrated, and the experimental schemes complement the picture of the work. Data demonstrating differences in the immune response to different vaccination options will undoubtedly be useful in the development of a vaccine against Chlamydia trachomatis. The important result of the work is the identification of vaccine-adjuvant combinations that provide effective protection against the pathogen. This result shows that the use of a cell-free membrane-bound protein synthesis strategy is a reliable basis for creating appropriate vaccines. The authors discussed the results obtained, and also reported on further studies needed to clarify the nature of the antigenic protein folding, which is necessary to transfer data from the mouse vaccine to the vaccine against Chlamydia trachomatis.
Overall, the work presents a well-formulated and experimentally confirmed platform for creating vaccines against membrane proteins that are difficult for vaccinology. The methods used in the work are not simple, but they all are reproducible. The undoubted advantage of the work is the clarity of presentation and explanation of all the nuances of the study.
I have no comments on the work; I consider it possible to publish the manuscript in its current form.
I would like to draw the authors' attention to the lack of information on the affiliation of the co-authors and I am surprised that the publishing system missed such an obvious oversight.
Author Response
REVIEW COMMENT 1: I would like to draw the authors' attention to the lack of information on the affiliation of the co-authors and I am surprised that the publishing system missed such an obvious oversight.
RESPONSE 1: We thank the reviewer for their detailed and constructive review of our manuscript Cell-free screening, production and animal testing of the major outer membrane protein of Chlamydia associated with STIs supported by nanolipoprotein. We appreciate your positive assessment of our work, including the experimental design, presentation of the MOMP-NLP complex assembly, and the discussion on vaccine-adjuvant combinations. We are pleased that the methods and findings are clear and relevant to advancing vaccine development against Chlamydia trachomatis.
Regarding the missing affiliations of some co-authors, this information is available on the revised manuscript and appreciate your attention to this matter. Thank you again for your thorough review and time regarding the review of our manuscript.
Reviewer 2 Report
Comments and Suggestions for Authors
The manuscript entitled "Cell-free screening, production and animal testing of STI-related chlamydial major outer membrane protein supported in Nanolipoproteins" represents a significant advance in the development of a vaccine against Chlamydia trachomatis. Although the research is thorough and well structured, several potential drawbacks and limitations need to be considered.
The authors report a significant reduction in Chlamydia muridarum infection in vaccinated mice, but the clinical relevance of these findings in terms of actual disease prevention in humans remains uncertain. Please comment.
The manuscript lacks a comparative analysis with existing vaccine candidates or approaches for Chlamydia. While the authors highlight the advantages of their MOMP-NLP formulation, a discussion of how it compares to other vaccine strategies, such as live attenuated or other subunit vaccines, would provide a more comprehensive understanding of its potential impact in the field.
The NLPs serve as a scaffold for MOMP, but the authors do not discuss in detail the potential limitations of this approach. For example, while NLPs can mimic the native environment of membrane proteins, the structural and functional integrity of MOMP within NLPs may not fully replicate its natural state in Chlamydia. This raises questions about the immunogenicity of the MOMP-NLP complex compared to native MOMP or MOMP presented in other formulations.
Comments on the Quality of English Language
Minor editing of English language required.
Author Response
REVIEW COMMENT 2.1: The manuscript entitled "Cell-free screening, production and animal testing of STI-related chlamydial major outer membrane protein supported in Nanolipoproteins" represents a significant advance in the development of a vaccine against Chlamydia trachomatis. Although the research is thorough and well structured, several potential drawbacks and limitations need to be considered.
RESPONSE 2.1: We thank the reviewer for their positive assessment of our manuscript and for recognizing our research in the development of a vaccine against Chlamydia trachomatis. We also appreciate the mention of potential drawbacks and limitations, and we acknowledge that no research is without its challenges.
In response, we have reviewed the manuscript to ensure these potential limitations are adequately and clearly discussed. Specifically, we have expanded the discussion to emphasize areas such as the translation of findings from animal models to human applications, the scalability of the production processes, and the need for further optimization of immune response to ensure long-term protection. We hope these additions within the manuscript clarify our approach to overcoming these challenges and highlight the promising aspects of our vaccine development strategy.
REVIEW COMMENT 2.2: The authors report a significant reduction in Chlamydia muridarum infection in vaccinated mice, but the clinical relevance of these findings in terms of actual disease prevention in humans remains uncertain. Please comment.
Response 2.2: We appreciate the reviewer’s comment regarding the clinical relevance of the Chlamydia muridarum infection reduction in mice and its applicability to human disease prevention in humans. In the introduction of our manuscript, we highlight the global public health burden posed by Chlamydia trachomatis. Although Chlamydia muridarum is a mouse-specific strain, it has widely been used as a model organism to study Chlamydia trachomatis due to its pathogenic similarities in the genital tract of mice and humans. Thus, significant infection reduction in mice serves as a meaningful preliminary step in understanding immune responses and advancing vaccine development.
Though the translation of animal model findings to humans poses challenges, ongoing research- including studies utilizing MOMP as a key antigen- demonstrates the potential for using the mouse model to develop an effective approach toward designing a Chlamydia trachomatis vaccine. By incorporating both our data and previous findings, we aim to bridge this gap and contribute to the development of a clinically relevant solution for Chlamydia prevention in humans. Our studies focus on identifying an approach to produce recombinant MOMP formulated with adjuvants that would be appropriate for use in humans studies. However, our studies focus on the internasal mouse model as it allows us to screen multiple formulations for down selection in follow on studies.
We hope this clarification provides a comprehensive response to the reviewer’s concerns, and we will continue to expand upon these efforts in our future work to ensure broader relevance in human disease prevention.
REVIEW COMMENT 2.3: The manuscript lacks a comparative analysis with existing vaccine candidates or approaches for Chlamydia. While the authors highlight the advantages of their MOMP-NLP formulation, a discussion of how it compares to other vaccine strategies, such as live attenuated or other subunit vaccines, would provide a more comprehensive understanding of its potential impact in the field.
Response 2.3: We would like to sincerely thank the reviewer for their valuable feedback and constructive criticism of our manuscript. We appreciate the insightful critique regarding the need for a comparative analysis with existing vaccines for Chlamydia.
In response to your suggestion, we have expanded our discussion to include a comprehensive comparison of our MOMP-NLP formulation with other vaccine strategies, specifically live attenuated vaccines and the structural limitations of subunit vaccines. Furthermore, we highlighted the difficulties in solubilizing full-length MOMP and how our NLP platform not only helps to maintain the structural integrity of the antigen, but also enhances its solubility and presentation to the immune system. The final advantage the NLP allows for quantitative incorporation of adjuvants as part of the vaccine formulation.
We believe that these changes within the discussion enhance the clarify and depth of our manuscript, providing a more comprehensive understanding of the potential impact of our MOMP-NLP formulation in the field of Chlamydia vaccine development, thank you once again for your thoughtful review.
REVIEW COMMENT 2.4: The NLPs serve as a scaffold for MOMP, but the authors do not discuss in detail the potential limitations of this approach. For example, while NLPs can mimic the native environment of membrane proteins, the structural and functional integrity of MOMP within NLPs may not fully replicate its natural state in Chlamydia. This raises questions about the immunogenicity of the MOMP-NLP complex compared to native MOMP or MOMP presented in other formulations.
Response 2.4: We appreciate your insightful comment regarding the limitations of using NLPs as scaffolds for MOMP. While NLPs can effectively mimic the native environment of membrane proteins, we acknowledge the concern whether the structural and functional integrity of MOMP within NLPs fully replicates its natural shape and form in Chlamydia. To address this, we included a discussion in the manuscript that outlines the potential discrepancies in immunogenicity between the MOMP-NLP complex and native MOMP, as well as MOMP presented in other vaccine formulations. Additionally, we highlighted the need for further structural analyses and comparative studies to better understand how these formulations might influence immune response and lead to better portection. Thank you for pointing out this important aspect.
Round 2
Reviewer 2 Report
Comments and Suggestions for Authors
The current version of the manuscript is suitable for publication in Vaccines.